# Analysis of early intermediate states of the nitrogenase reaction by regularization of EPR spectra

Lorenz Heidinger[1,2], Kathryn Perez[3], Thomas Spatzal[3], Oliver Einsle[2], Stefan Weber [1], Douglas C. Rees [3] ✉ & Erik Schleicher [1] ✉

Due to the complexity of the catalytic FeMo cofactor site in nitrogenases that mediates the reduction of molecular nitrogen to ammonium, mechanistic details of this reaction remain under debate. In this study, selenium- and sulfur-incorporated FeMo cofactors of the catalytic MoFe protein component from *Azotobacter vinelandii* are prepared under turnover conditions and investigated by using different EPR methods. Complex signal patterns are observed in the continuous wave EPR spectra of selenium-incorporated samples, which are analyzed by Tikhonov regularization, a method that has not yet been applied to high spin systems of transition metal cofactors, and by an already established grid-of-error approach. Both methods yield similar probability distributions that reveal the presence of at least four other species with different electronic structures in addition to the ground state $E_0$. Two of these species were preliminary assigned to hydrogenated $E_2$ states. In addition, advanced pulsed-EPR experiments are utilized to verify the incorporation of sulfur and selenium into the FeMo cofactor, and to assign hyperfine couplings of $^{33}S$ and $^{77}Se$ that directly couple to the FeMo cluster. With this analysis, we report selenium incorporation under turnover conditions as a straightforward approach to stabilize and analyze early intermediate states of the FeMo cofactor.

The conversion of the largely inert $N_2$ molecule to bioavailable ammonia is essential for life on Earth and is a critical step in the biological nitrogen cycle. Biological nitrogen fixation is catalyzed by enzymes of the nitrogenase family that are widespread in bacteria and archaea, but absent in eukaryotes[1]. Three isoforms of nitrogenases are distinguished based on the composition of their catalytic cofactor: the Mo-dependent, V-dependent, and Fe-only nitrogenases[2,3]. All nitrogenases are two-component proteins consisting of (i) the [4Fe:4 S] cluster-containing homodimeric Fe-protein (component Av2 in *Azotobacter vinelandii*) that serves as reductase and site of ATP hydrolysis and (ii) the catalytic, $\alpha_2\beta_2$-heterotetrameric (or heterohexameric in case of V and Fe) MoFe protein (component Av1 in *Azotobacter vinelandii*) with two metal cofactors, the [8Fe:7 S] P-cluster and the catalytic cofactor. The latter is designated as FeMo cofactor in Mo-dependent nitrogenases and is the most complex bioinorganic metal cluster known to date. The FeMo cofactor consists of seven Fe atoms, nine S atoms, one Mo atom, a central C (carbide) atom, and an organic R-homocitrate moiety (Fig. 1, inset), and is accordingly complex in its electronic and magnetic properties[4–7].

Important traits of the molecular mechanism of nitrogen reduction remain under discussion. It is known that the FeMo cofactor binds the natural substrate $N_2$ (alternatively also a variety of other small

[1]Institut für Physikalische Chemie, Albert-Ludwigs-Universität Freiburg, Freiburg, Germany. [2]Institut für Biochemie, Albert-Ludwigs-Universität Freiburg, Freiburg, Germany. [3]Howard Hughes Medical Institute (HHMI), California Institute of Technology, Division of Chemistry and Chemical Engineering, Pasadena, CA, USA. ✉e-mail: dcrees@caltech.edu; erik.schleicher@pc.uni-freiburg.de

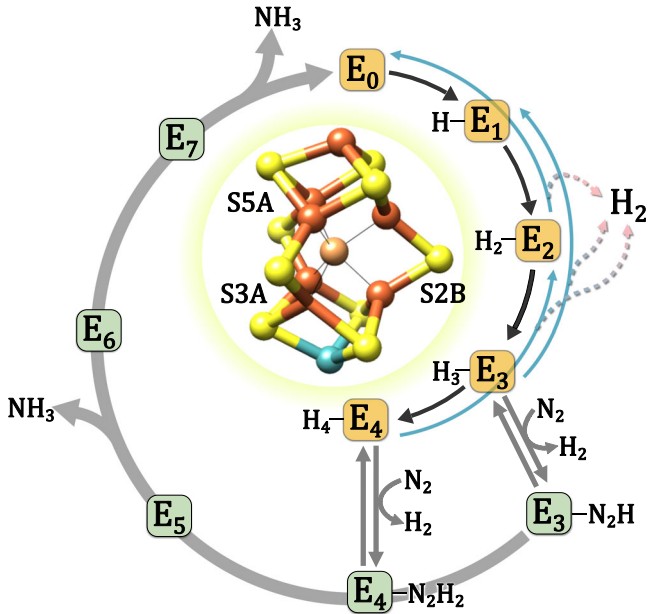

**Fig. 1 | Lowe-Thorneley model for nitrogenase.** The reaction cycle postulates an eight-electron process and consequently proceeds through eight different one-electron steps ($E_0$–$E_7$), assuming an alternating transfer of electrons and protons. The binding of the substrate $N_2$ occurs in the $E_3$ or $E_4$ states. While nonproductive $H_2$ generation is observed in the $E_0$–$E_4$ states (blue lines), the exchange of $N_2$ for $H_2$ is a mechanistic requirement. Inset: Molecular structure of the FeMo cofactor. Iron and sulfur atoms of the cofactor are colored according to standard nomenclature, Mo is shown in blue and the central C in beige (structure is generated from PDB entry 4TKU[26]). Selected ring sulfur atoms are additionally labeled.

molecules such as CO) during catalysis and strictly sequentially accepts electrons from the [4Fe:4 S] cluster of the Fe protein. This transfer is coupled to the hydrolysis of 2 ATP/e$^-$ by the Fe protein, whereby one electron is first transferred from the reduced P cluster to the FeMo cofactor, and the electron deficit at the P cluster is subsequently replenished by the Fe protein[8]. The reductase component then dissociates from the MoFe protein for reduction and nucleotide exchange before the next 1-electron transfer can take place[9]. Largely due to the complexity of this process, Fe protein is the only known reductant to sustain productive $N_2$ reduction by MoFe protein, although recent electrochemical approaches have been reported to achieve similar results[10]. The reduction of $N_2$ follows a minimal stoichiometry of

$$N_2 + 8\,e^- + 8\,H^+ + 16\,ATP \rightarrow 2\,NH_3 + H_2 + 16\,[ADP + P_i]$$

including the obligatory release of $H_2$ with a limiting stoichiometry of $1\,H_2/N_2$. The kinetics of the reaction are comprehensively outlined in a scheme proposed by Lowe and Thorneley (LT)[11], in which the system cycles through eight distinct states, $E_0$ to $E_7$, each representing the addition of a single electron (Fig. 1). Under reductive conditions the FeMo cofactor is commonly isolated in the resting state $E_0$[11], and then successively receives electrons (and protons) through states $E_1$ to $E_7$. Importantly, the binding and activation of $N_2$ requires the enzyme to reach state $E_3$ or $E_4$, which is complicated by the risk of an unproductive loss of 2 electrons as additional $H_2$[2,12]. This finding indicated that an essential aspect of electron accumulation on the FeMo cofactor is the formation of surface hydrides that can be lost as $H_2$ by accidental protonation[3,13]. Stabilization of these surface-associated hydride adducts may be achieved by a bridging binding mode[14]; this type of electron storage is crucial for the cluster to accumulate four electrons at isopotential (i.e., from the Fe protein) and allows for a mechanistic twist upon reaching the $E_3$ or $E_4$

state. Triggered by the presence or binding of the substrate $N_2$, the two adjacent hydrides present in the $E_4$ state can reductively eliminate $H_2$, leaving the enzyme in a 2-electron-reduced state that cannot be achieved by electron transfer from the Fe protein alone, and that is sufficiently reactive to break the $N_2$ triple bond[15]. From states $E_5$–$E_7$, the reaction then proceeds to the release of the product $NH_3$, but different mechanistic routes remain under debate[2,16,17].

The $E_0$ state of the FeMo cofactor has a total spin of $S = 3/2$[18,19] and the oxidation state of the FeMo cofactor changes by 1 with each reaction step; so that the total spin of the cofactor is half-integer for any even state and integer for any odd state. The odd states are thus either diamagnetic ($S = 0$) or have non-Kramers spin states[2] ($S = 1, 2, 3, ...$) with high zero-field splitting and hence the absence of EPR transitions at common EPR frequencies[20,21]. EPR spectroscopy provides access to the characterization of the ground state as well as the $S \neq 0$ reaction intermediates and supports the drawing of mechanistic and – within limits – also structural conclusions. In particular, freeze-quenched samples with different substrates, some of them stable-isotope-labeled, have been studied[22–24]. Several of these studies showed complex continuous-wave (cw)-EPR spectra with well-resolved anisotropy of the *g*-tensor, indicating that the substrate directly couples with at least one Fe atom of the FeMo cofactor[24,25]. However, an unambiguous assignment of the binding position was not possible.

The substrate binding site of the CO-inhibited FeMo cofactor in its resting state was identified by crystallography[26,27]. CO displaces the S at position S2B and a CO bond in an end-on $\mu_2$-bridging mode to Fe2 and Fe6 is formed at this position[26,27]. In a subsequent study, KSeCN was found to be both a substrate and an inhibitor of nitrogenase activity and crystal structures from freeze-quenched nitrogenase samples generated during turnover with KSeCN revealed that S2B was replaced by Se[28]. When KSeCN was removed from the reaction mixture and the reaction was allowed to proceed, further Se exchange first occurred at positions 3 A and 5 A, the other two $\mu_2$-bridging S that form the equatorial 'belt' of the cofactor (Fig. 1, inset). Only after several thousand more reaction cycles the incorporated Se was again replaced by S. Starting from the exclusive Se2B labeling, the approximately equal labeling distribution of the other two positions (3 A and 5 A) was reached after about 1000 turnover cycles[28]. Comparable S-to-S exchange experiments within the sulfur belt were carried out with the VFe cofactor of V-dependent nitrogenase[29]. A subsequent study examining Se incorporation into the FeMo cofactor of a Mo-dependent nitrogenase at high and low KSeCN concentrations established that both conditions lead to a similar Se distribution within the cofactor[30]. Furthermore, it could be demonstrated that Se labeling is also possible at positions 3 A and 5 A by gassing the 2B-Se-labeled protein with CO during catalysis. In this process, the Se2B is exchanged by CO, while the two S atoms at the 3 A/5 A positions are replaced by Se. The use of such a Se-labeled FeMo cofactor allowed its electronic structure to be analyzed by various methods like X-ray spectroscopy[30].

This work analyzes whether and to what extent Se is incorporated into the FeMo cofactor and what geometric or electronic changes result from this manipulation. We use high-resolution EPR spectroscopy for this purpose, as structure-determination methods can identify the labeling positions of individual isotopes within the FeMo cofactor, but the various electronic structures or redox states of the cluster are difficult to be distinguished other than with complex approaches like spatially resolved anomalous dispersion refinement[31]. Tikhonov regularization, commonly applied to analyze complex magnetic resonance datasets, e.g., from PELDOR/DEER spectroscopy[32–36], was employed on cw-EPR spectra of the high-spin FeMo cofactor to assign individual species formed by Se incorporation. The resulting probability distributions revealed several species with different electronic structures in each sample, making

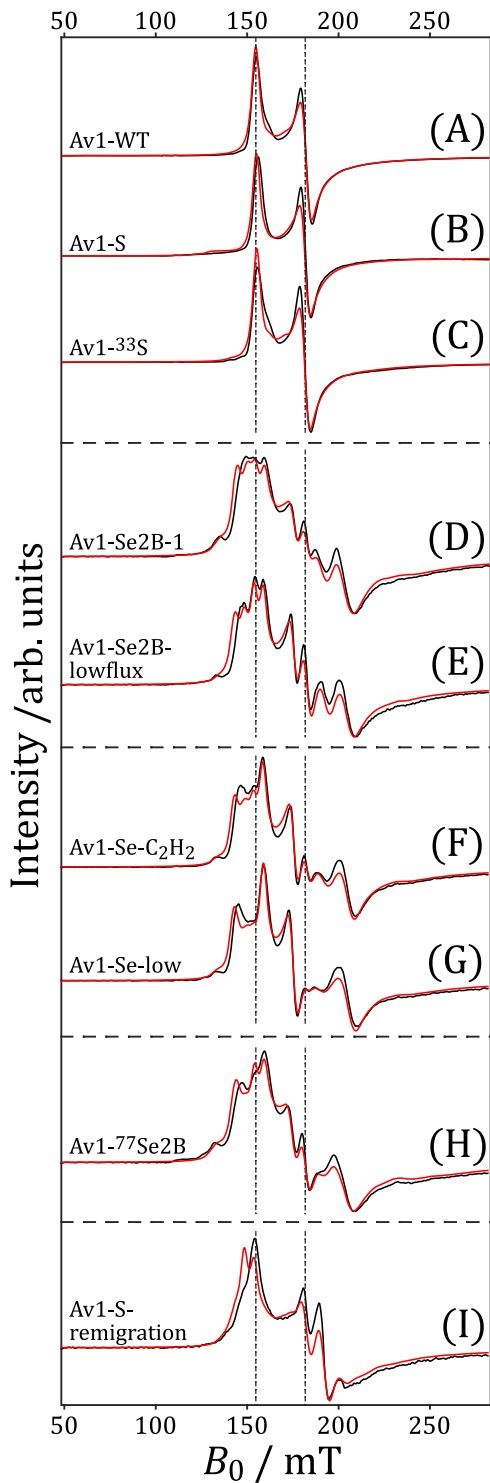

**Fig. 2 | EPR spectroscopy of Se-incorporated Av1 samples.** Normalized baseline-subtracted X-Band cw-EPR spectra (black) of samples Av1-WT (**A**), Av1-S (**B**), Av1-$^{33}$S (**C**), Av1-Se2B-1 (**D**), Av1-Se2B-lowflux (**E**), Av1-Se-$C_2H_2$ (**F**), Av1-Se-low (**G**), Av1-$^{77}$Se2B (**H**), and Av1-S-remigration (**I**), measured with a microwave power of 37.7 mW at $T = 5$ K. Calculated spectra obtained from regularization using a linewidth of 2.5 mT are depicted in red. Dashed vertical lines depict two principal $g$-values of Av1-WT. Full-range cw-EPR spectra covering the magnetic field range of 50–400 mT are depicted in Supplementary Fig. 16.

an assignment to specific intermediates and/or redox states possible. The quality of our analyses was compared to those obtained from a grid-of-error approach[37]. Together, these studies establish that Se incorporation into the FeMo cofactor provides access to

other states in the kinetic LT scheme that will help to better understand the molecular mechanism of the FeMo cofactor in the nitrogenase reaction.

## Results

### Regularization of cw-EPR spectra

To spectroscopically follow the changes of the FeMo cofactor after incorporation of Se, different nitrogenase Av1 samples were produced under various turnover conditions in the absence of $N_2$ (see Enzyme assays in the Method section); these samples exhibited different labeling positions (position 2B and/or positions 2B, 3 A, and 5 A) or labeling yields. For this purpose, different KSeCN and KSCN concentrations (samples Av1-Se2B-1, Av1-Se-low, Av1-Se2B-lowflux), different Av1/Av2 ratios (samples Av1-Se2B-lowflux and Av1-S) and different reaction cycles (samples Av1-Se-$C_2H_2$ and Av1-S-remigration) were applied. Two S-incorporated samples, one with $^{33}$S (Av1-$^{33}$S) and one with natural abundance $^{32}$S (Av1-S) were prepared under turnover conditions and analyzed in comparison. All samples were frozen after the defined number of reaction cycles but not under freeze-quench conditions. Therefore, no short-lived intermediate states are expected to be trapped. Figure 2 depicts the cw-EPR spectra of all Av1 samples under investigation covering a magnetic field range of 50–283 mT.

The Av1-WT sample in its resting state exhibits the well-known $S = 3/2$ spin state EPR spectrum of the lower Kramer's doublet (panel A). The two EPR spectra of the S-incorporated samples, Av1-S and Avl-$^{33}$S (panels B and C) are virtually identical compared to the unmodified protein; therefore, incorporation of S and in particular $^{33}$S (with a nuclear spin of $I = 3/2$) into the FeMo cofactor is not detectable by cw-EPR spectroscopy. All Se-exchanged samples, however, exhibit a complex signal shape with at least five signals spanning the 120 – 260 mT magnetic field range. Unexpectedly, the "Se-patterns" of samples Av1-Se2B-1, Av1-Se2B-lowflux, Av1-Se-$C_2H_2$, Av1-Se-low, and even of Avl-$^{77}$Se2B (panels D–H) are similar, only differences in individual peaks intensities can be observed. It is important to note that $^{77}$Se has a nuclear spin of $I = \frac{1}{2}$, which is different to the $I = 0$ of the naturally most abundant isotopes $^{78}$Se and $^{80}$Se. As those samples show very similar spectral patterns, hyperfine couplings of $^{77}$Se and the FeMo cofactor can be excluded as the origin of the Se-pattern. The cw-EPR spectrum of the Av1-S-remigration sample (panel I) again exhibits the Se-pattern, but with decreased intensity. Qualitatively, the observed signal pattern can be described as a mixture of signals from unlabeled and Se-incorporated samples.

For a more quantitative evaluation of S-, Se- and unlabeled samples, the intensity differences of the respective cw-EPR spectra were compared using spin counting via double integration. Samples Av1-WT, Av1Se2B-1, Av1-$^{77}$Se2B, Av1-Se-low, Av1-Se-$C_2H_2$, and Av1-$^{33}$S were compared, as all were prepared from the same enzyme batch and under identical electron flux. The analysis shows that the signal intensity of sample Av1-$^{33}$S is comparable to the intensity of the Av1-WT sample, but all Se-incorporated samples have only ≈60% of the resting-state intensity (Supplementary Fig. 17). Consequently, Se incorporation leads to ≈40% EPR-inactive ($S = 0$) and/or non-Kramers states ($S = 1, 2, 3,...$).

It is essential to know the origin of the complex Se-pattern to perform correct spectral simulations of the experimental data. Hyperfine couplings have already been ruled out as the source, geometric distortions due to Se incorporation are also unlikely as the only explanation, as there is no evidence for such in the crystal structures[28], assuming that the Se incorporation in crystals is representative of that in solutions. Moreover, the EPR signal pattern of sample Av1-Se-$C_2H_2$, in which Se should be incorporated over the entire sulfur belt, is almost identical to those of the other Se-incorporated samples labeled mainly at the 2B position (see also below). Therefore, different states of the FeMo cofactor that manifest in different zero-field splitting parameters

are the most plausible assumptions. In this case, the cw-EPR spectra of all samples are dominated only by the rhombicity parameter ($\lambda$) of the zero-field splitting as the effective $g$-factors $g_{\{x,y,z\}}^{1/2}$ of the lower Kramer doublet of an $S = 3/2$ system are functions of $\lambda = |E/D|$ (see Supplementary "Methods").

Exact $|E/D|$ values are thus desired for a precise simulation of pulsed EPR data as the zero-field Hamiltonian $H_{ZFS}$ depends on $|D|$ and $\lambda = |E/D|$. $|D|$ can be estimated experimentally by temperature-dependent measurements of the intensity ratios of the lower and upper Kramers doublet at $g \approx 6$[38]. These measurements were conducted on samples Av1-WT and Av1-Se2B-1 at 6 K and 15 K (Supplementary Fig. 18), and small differences were observed: The signal of the latter sample is slightly shifted to ≈115 mT and shows a more complex signal pattern compared to the single signal at 111 mT in the Av1-WT sample. However, quantitative extraction of signal intensities was not possible due to the substantial overlap of the signals from the lower and upper Kramer doublet (Supplementary Fig. 18). Nevertheless, the analysis demonstrates that $|D|$ (and the effective $g$-factors) is of the same magnitude in the Se-incorporated samples. Please note that the effective $g$-values are independent of $D$, if the energy of the Zeeman interaction is small compared to zero-field energy.

Inhomogeneous broadening of the magnetic parameters of protein-bound (metal) cofactors is usually approximated by a random distribution of the EPR parameters, in particular the $\boldsymbol{D}$- and $\boldsymbol{g}$-tensors, using Gaussian distributions, so-called strain models[39–42]. These distribution models are valid as long as the width of the distribution is small compared to its magnitude. However, the experimental spectra of the high-spin Se-FeMo cofactor exhibit a large splitting compared to their size (Fig. 2), so such simple strain models cannot correctly reproduce these data sets, and thus other approaches are required.

Having only the parameter $\lambda$ that dominates the cw-EPR spectrum, a regularization method was applied to deconvolute the complex signal pattern in the Se-incorporated samples (see Supplementary "Methods" for theoretical details). Briefly, ill-posed problems can be solved by Tikhonov regularization. First, the potential and robustness of the regularization method was thoroughly tested using three calculated model datasets (Supplementary Table 1). After the optimal regularization parameter $\alpha_{Opt}$ was determined by different methods, the distribution function was obtained. From this, the respective cw-EPR spectrum was calculated (Supplementary Fig. 3–12). The regularization reproduced the calculated model spectra very well (Supplementary Fig. 9–12), and therefore, the method was used to analyze all experimental Av1 cw-EPR spectra. As a common regularization considers one dominant parameter ($\lambda$), an intrinsic linewidth (lwpp) analysis of all samples was first performed, and optimal intrinsic Lorentzian peak-to-peak line shapes of 2.5–3 mT, 3.0–3.5 mT, and 3.5–4.0 mT were obtained for spectra recorded at 5–6 K, 9 K, and 12 K, respectively (Supplementary Methods and Supplementary Fig. 19–23). Moreover, the real principal $g$-values of all species were assumed to be identical.

The distribution functions obtained from regularization are shown in Fig. 3, and the individual $\lambda$ values of all species are summarized in Table 1. A multi-Gaussian fit was applied to quantify the individual distributions (Supplementary Fig. 26 and Supplementary Table 2). It is observed that samples Av1-WT, Av1-S, and Av1-$^{33}$S (panels A–C) contain only one spin species with an average value of $\lambda_2 = 0.054$. In contrast, samples Av1-Se2B-1, Av1-Se2B-lowflux, Av1-Se-C$_2$H$_2$, Av1-Se-low and Av1-$^{77}$Se2B (panels D–H) contain five species with average $\lambda$ values of $\lambda_1 = 0.033$, $\lambda_2 = 0.057$, $\lambda_3 = 0.082$, $\lambda_4 = 0.116$ and $\lambda_5 \approx 0.19$. The second value, $\lambda_2$, matches that of the Av1-WT and accordingly was assigned to the electronic resting state of the FeMo cofactor (E$_0$). Even though the other four "Se-species" are present in all Se-incorporated samples, noticeable population differences between samples can

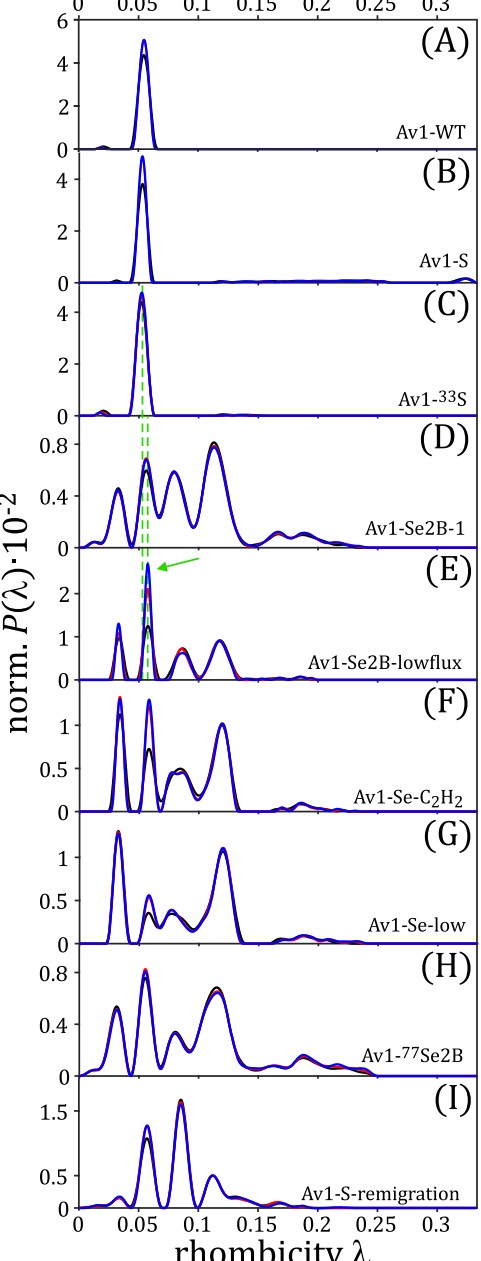

**Fig. 3 | Analysis of EPR spectra via regularization.** Normalized probability distributions P($\lambda$) obtained by regularization of cw-EPR spectra (microwave powers: 37.7 mW (black), 3.77 mW (red) and 0.377 mW (blue)). An lwpp of 2.5 mT was used. The samples are as follows: (**A**) Av1-WT, (**B**) Av1-S, (**C**) Av1-$^{33}$S, (**D**) Av1-Se2B-1, (**E**) Av1-Se2B-lowflux, (**F**) Av1-Se- C$_2$H$_2$, (**G**) Av1-Se-low, (**H**) Av1-$^{77}$Se2B, (**I**) Av1-S-remigration. Green dashed vertical lines illustrate the differences of species $\lambda_2$ between samples with and without Se-incorporation.

be detected. In Av1-Se2B-1 and Av1-$^{77}$Se2B, all four Se-species are populated, with $\lambda_4$ being the largest fraction ( ~ 36%, black triangles in Table 1). In Av1-Se-low, on the other hand, the fraction of species $\lambda_2$ is below 10%, the Se-species are more highly populated, in particular $\lambda_4$. It is worth noting that the $\lambda$ populations of samples Av1-Se2B-1 and Av1-Se2B-lowflux differ; in contrast to Av1-Se2B-1, sample Av1-Se2B-lowflux shows predominantly $\lambda_2$ and only small amounts of any of the Se-species. This can be rationalized by a lower electron flux in sample Av1-Se2B-lowflux due to the lower Av2/Av1 ratio, which in turn might result in a decreased formation rate of Se-species per time. The largest $\lambda_5$ value of ≈0.19 has a very broad $\lambda$ distribution and in most cases only a

**Table 1 | Summary of rhombicity parameters (extracted from Fig. 3)**

| | | $\lambda_1$ (fraction %) | $\lambda_2$ (fraction %) | $\lambda_3$ (fraction %) | $\lambda_4$ (fraction %) | $\lambda_5$ (fraction %) |
|---|---|---|---|---|---|---|
| Av1-WT | | | 0.055 | | | |
| Av1-S | | | 0.053 | | | |
| Av1-33S | | | 0.053 | | | |
| Av1-Se2B-1 | | 0.033 (11.5%) | 0.056 (17.2%) | 0.080 (23.3%) | 0.113 (35.9%) | 0.166–0.19 (12.1%▲) |
| Av1-Se2B-lowflux | | 0.033 (14.4%) | 0.058 (34.3%▲) | 0.086 (18.4%) | 0.118 (30%) | ≈0.189 (2.9%) |
| Av1-$^{77}$Se2B | | 0.032 (14.1%) | 0.056 (20.2%) | 0.080 (10.6%) | 0.115 (37.6%) | 0.163–0.220 (4.3%) |
| Av1-Se-low | | 0.033 (24.2%▲) | 0.058 (8.4%) | 0.078 (22%) | 0.120 (38.9%▲) | 0.171–0.230 (4.3%) |
| Av1-Se-C$_2$H$_2$ | | 0.034 (19.2%) | 0.058 (20.8%) | 0.084 (10.6%) | 0.120 (33.4%) | 0.175–0.217 (6.5%) |
| Av1-S-remigration | | 0.033 (4.1%) | 0.057 (30.5%) | 0.086 (39.4%▲) | 0.112 (16.1%) | 0.180 (9.9%) |
| Average value | | 0.033 | 0.056 | 0.082 | 0.116 | ≈0.19 |
| Effective $g$-values[a] | $g'^{1/2}_x$ | 3.80 | 3.66 | 3.50 | 3.28 | ≈2.82 |
| | $g'^{1/2}_y$ | 4.20 | 4.33 | 4.48 | 4.64 | ≈4.98 |
| | $g'^{1/2}_z$ | 2.02 | 2.01 | 1.99 | 1.95 | ≈1.83 |

[a] Calculated from Supplementary Methods Equation 3 and $g_x = g_y = 2.00$ and $g_z = 2.03$.

$\lambda$ values were extracted manually by determining the local maxima of the distribution, values with a probability height of less than 10% were ignored from further discussion. The respective populations of all Se-incorporated samples were fitted using a multi-Gaussian function (Supplementary Table 2), and all fractions were determined. The respective largest fraction is marked with a black triangle (▲). Effective $g$-factors were calculated by using Equation 3 (Supplementary Methods).

low (<10%) population. Sample Av1-S-remigration (panel I), in which the Se is expected to be re-replaced by S, shows a different distribution than any of the other Se-incorporated samples: Species $\lambda_1$ and $\lambda_4$ are depopulated, and in addition to the resting state, only the $\lambda_3$ state is populated.

To evaluate the relaxation behavior of the individual spin species, cw-EPR spectra were recorded at different microwave powers of 0.377 mW, 3.77 mW, and 37.7 mW for analysis by regularization (Fig. 3, red and blue lines, additional microwave powers are shown as Supplementary Fig. 24). The relaxation behavior of all Se-species is similar, but different from that of the resting-state FeMo cofactor ($\lambda_2$). Temperature-dependent measurements at 6, 9, and 12 K produced similar results (Supplementary Fig. 20–23).

From the normalized population distributions (Fig. 3), cw-EPR spectra were calculated (red lines in Fig. 2). The agreement between experiment and regularization is remarkably good in all samples and demonstrates the potential of the regularization method. Slight differences, e.g., in the signals at 145 mT and 200 mT (panels D–I), are only intensity differences and are most likely caused by small baseline artifacts.

**Analysis of cw-EPR spectra using the grid-of-error method**

The question remains whether the cw-EPR spectra are dominated only by the $\lambda$ parameter or whether the intrinsic line shape lwpp is a second important parameter that differs between samples and/or between individual spin species. Therefore, the established grid-of-error approach[37] was used as a second method to re-evaluate all Av1 cw-EPR spectra. The results are depicted in Fig. 4 and demonstrate that this method yields similar distribution functions compared to the regularization method. It is noteworthy that the $P(\lambda)$ functions are significantly narrower than those obtained by regularization. This is not surprising, as the width of the distribution is partially compensated by a distribution of the intrinsic spectral linewidths. Again, samples Av1-WT Av1-S

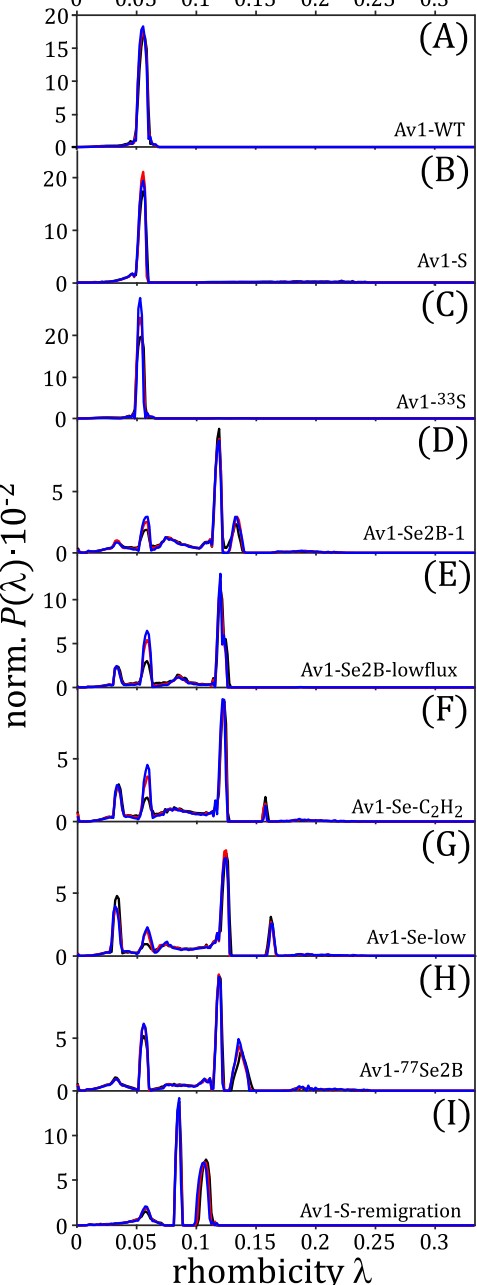

**Fig. 4 | Analysis of EPR spectra via Grid-of-Errors.** Normalized probability distributions P($\lambda$), calculated from the full linewidth distribution graphs $P(\lambda, \text{lwpp})$ (Supplementary Fig. 25) by summation over all lwpp and subsequent normalization. Different microwave powers are shown as black (37.7 mW), red (3.77 mW), and blue (0.377 mW) curves, respectively. The samples are as follows: (**A**) Av1-WT, (**B**) Av1-S, (**C**) Av1-$^{33}$S, (**D**) Av1-Se2B-1, (**E**) Av1-Se2B-lowflux, (**F**) Av1-Se- C$_2$H$_2$, (**G**) Av1-Se-low, (**H**) Av1-$^{77}$Se2B, (**I**) Av1-S-remigration.

and Av1-$^{33}$S (panel A–C) contain only one species with a $\lambda = 0.054$ value, and samples Av1-Se2B-1, Av1-Se2B-lowflux, Av1-Se-C$_2$H$_2$, Av1-Se-low and Av1-$^{77}$Se2B (panels D–H) contain four Se-species with $\lambda$ values of $\lambda_1 = 0.035$, $\lambda_2 = 0.058$, $\lambda_3 = 0.085$, $\lambda_4 = 0.12$. A fifth species with a $\lambda$ value of around ≈ 0.19 can be detected in samples Av1-Se2B-1, Av1-Se-C$_2$H$_2$, Av1-Se-low, and Av1-$^{77}$Se2B. Sample Av1-S-remigration (panel I) shows only three species with $\lambda$ values of 0.058, 0.085, and 0.12. These $\lambda$ values are very similar to those obtained by regularization.

Qualitatively, both methods yield similar population trends for all Se-incorporated samples. However, the individual populations

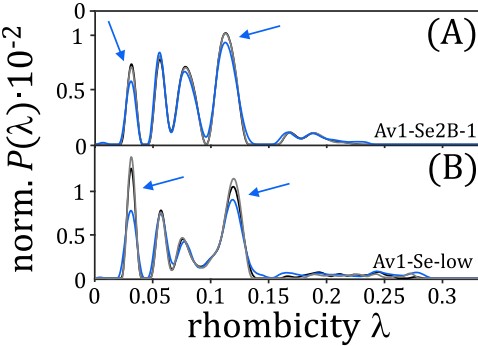

**Fig. 5 | Analysis of low-temperature photolysis experiments.** Normalized probability distributions $P(\lambda)$ obtained by regularization of cw-EPR spectra of samples Av1-Se2B-1 (**A**) and Av1-Se-low (**B**). Spectra are recorded at 6 K in the dark (black), after 10 min of blue light illumination (light blue), and after cryo-annealing at 150 K in the dark (grey).

differ depending on the method of analysis, and as we believe that the regularization provides more reliable populations, only for this method, a quantitative evaluation was carried out (Table 1). One major advantage of the grid-of-error method is that two (or even more) parameters can be optimized simultaneously so that line-widths are obtained for all species analyzed. A 2-dimensional representation ($\lambda$ and lwpp) shows that the non-Se-incorporated cofactors exhibit a lwpp between 1 mT and 3 mT (Supplementary Fig. 25), consistent with the result of 2.5 mT from regularization. The analysis of the Se-incorporated samples confirms that the lwpp of $\lambda_1$, $\lambda_2$ and $\lambda_3$ are between 1–3 mT, and only the lwpp of $\lambda_4$ is significantly larger than 5 mT. This result is unexpected, as the analyses of the relaxation times led to similar values for all Se-incorporated samples (see below). One explanation could be that the width of the individual $\lambda_4$ values is significantly broader than $\lambda_{1-3}$, mainly because the grid-of-error method tends to overrate the parameter lwpp (see also section Regularization versus grid-of-error approach).

## Light excited experiments

Hoffman and coworkers[14] have used intra-EPR cavity photolysis at 450 nm to characterize hydride-containing states of the FeMo cofactor; by irradiating nitrogenase samples with blue light and subsequent annealing at 150 K, a conversion of two $E_2(2H)$ isomers (denoted as 1b and 1c) could be demonstrated. Following these studies, samples Av1-Se2B-1 and Av1-Se-low were used to perform such experiments. The respective cw-EPR spectra (Supplementary Fig. 27) were analyzed by regularization and are shown in Fig. 5. It is evident that both samples respond to light irradiation and subsequent cryo-annealing, i.e., the probability distributions of the species change, but the changes are more pronounced in sample Av1-Se-low. This may be due to the fact that this sample contains the lowest fraction of state $E_0$.

In contrast to the results presented in reference[14], no species interconvert upon light illumination, but rather only a reduction of signal intensities can be detected (blue arrows in Fig. 5). A one-to-one correspondence to the published results cannot be expected, however, as the FeMo cofactor used in reference[14] and the Se-FeMo cofactors and accompanying intermediates studied in our experiments do have slightly different properties such as binding strengths and absorption coefficients. The regularization clearly shows that the population probabilities of the individual species are different: while $\lambda_2$ and $\lambda_3$ do not change, the population probabilities of $\lambda_1$ and $\lambda_4$ decrease significantly, and similarly. As the ground state $\lambda_2$ is not supposed to change, we can identify two distinct responses: The population probabilities of $\lambda_1$ and $\lambda_4$ change with light, those of $\lambda_3$ do not.

## Pulse EPR experiments

Prior analyses of hyperfine couplings, transient nutation, inversion recovery, and 2-pulse ESEEM experiments were conducted at Q-band microwave frequencies to determine the relaxation times and spin states of all samples. The transient nutation experiments revealed that unlabeled and Se-incorporated samples contain the same nutation frequencies, and only the intensities and linewidths of individual signals differ to a small extent (Supplementary Fig. 28). Therefore, all "Se-species" must possess the same total spin as the FeMo cofactor in its resting state ($S = 3/2$). Analysis of 2-pulse ESEEM and inversion recovery spectra yielded the relaxation times $T_M^{eff}$ and $T_1^{eff}$, which are in the range of 200–400 ns and 1–3 μs, respectively (Supplementary Fig. 29 and 30). The relaxation times of all samples are similar and are too short to conduct certain pulse experiments like ENDOR spectroscopy under our experimental conditions.

Representative Q-Band $\tau$-averaged 2-pulse ESEEM experiments of samples Av1-WT and Av1-Se2B-1 are depicted as upper panels of Fig. 6. Additionally, the pseudo-modulated spectra are shown for a direct comparison with the cw-EPR spectra in X-band shown in Fig. 2 A/D. Both spectra are quite similar to the ones obtained from X-band microwave frequencies: the Av1-WT sample shows the typical spectrum of the FeMo cofactor in its resting state (Fig. 6, left), and the Av1-Se2B-1 sample shows the already described complex Se-pattern (Fig. 6, right). However, the signal-to-noise ratio (S/N) of the pulse EPR spectrum is significantly lower, which is mainly due to the lock-in detection of the cw-EPR spectra, and the intensities of the individual signals differ slightly due to the incomplete compensation of different ESEEM modulation depths at different magnetic field positions by $\tau$-averaging.

3P-ESEEM spectra (Fig. 6, lower panels) of Av1-WT (black traces), Av1-Se2B-1 (red traces), and Av1-S (dark blue traces) are depicted at four different magnetic-field positions (580, 660, 740, and 880 mT), these spectra show nearly identical hyperfine couplings close to the proton Larmor frequency and in the range between 0–5 MHz; the latter signals have been assigned to two nitrogen atoms of the surrounding amino acids[43,44]. Using literature values[43,44], the ESEEM signals of the three samples can be simulated with good agreement. This result confirms that the direct protein environment of the FeMo cofactor remains structurally intact after turnover with KSeCN, and that no other ligand such as SeCN⁻ or CN⁻ is attached to the cluster. In addition, it is reconfirmed that the overall spin of the cluster remains the same; otherwise, additional nitrogen hyperfine couplings would be expected.

On the other hand, samples Av1-77Se2B (orange traces) and Av1-33S (light blue traces) show additional resonances (shaded orange and light blue areas in Fig. 6), which originate from hyperfine couplings of the respective EPR-active nuclei (33S and 77Se) and the FeMo cofactor. Differences in the frequencies and signal patterns are due to different Larmor frequencies of the two nuclei and additional quadrupole couplings in the case of 33S. Sample Av1-Se2B-1 does not show any Se hyperfine couplings as the natural abundance of 77Se is below 8%. Spectral simulations of these additional hyperfine couplings are required for a quantitative analysis. However, such simulations are complex because at almost all magnetic positions the EPR spectra of the Se-species overlap, and therefore the observed 77Se hyperfine couplings are the weighted sum of each species' contribution.

Additional difficulties arise when simulating the 33S hyperfine couplings in sample Av1-33S, as the quadrupole coupling of the 33S nucleus overlaps strongly with the resonances of the two 14N nuclei. Moreover, depending on the magnetic-field position, different ESEEM resonances are suppressed due to cross-suppression effects, and the 3P-ESEEM spectrum of two 14N nuclei and one 33S nucleus shows a large number of peaks due to the product rule. Therefore, no unequivocal spectral simulation could be achieved. Qualitatively, the few signals in

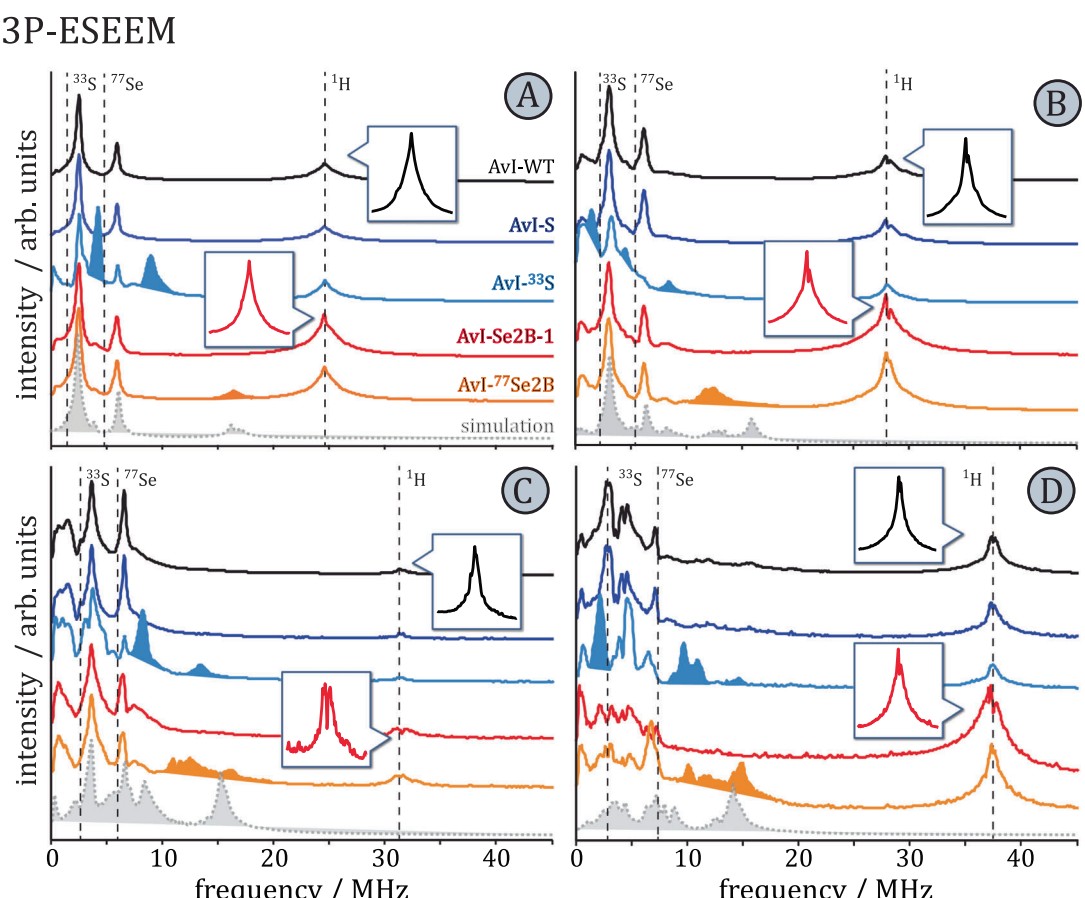

**Fig. 6 | Pulse Q-band EPR spectroscopy.** Upper panel: τ-averaged echo-detected and pseudo-modulated spectra of Av1-WT (left) and Av1-Se2B-1 (right). Grey arrows indicate the magnetic-field positions at which 3P-ESEEM experiments are recorded (**A**: 580 mT, **B**: 660 mT, **C**: 740 mT and **D**: 880 mT). Lower panels: 3P-ESEEM experiments of Av1-WT (black), Av1-Se2B-1 (dark blue), Av1-$^{77}$Se2B (light blue), Av1-S (red) and Av1-$^{33}$S (orange). Shaded areas highlight selected differences in the signal patterns as compared to the Av1-WT sample. Spectral simulations of Av1-$^{77}$Se2B are shown as dotted grey lines. Insets show expansions of the region around the proton Larmor frequency. Additional 3P-ESEEM experiments measured at different magnetic-field positions are summarized in Supplementary Fig. 31.

the 580 mT and 660 mT spectra indicate that a single $^{33}$S nucleus with hyperfine and quadrupole couplings of a few MHz can generate such a pattern.

Using published $^{14}$N hyperfine couplings and assuming one $^{77}$Se nucleus, the analysis of the spectral pattern in the Av1-$^{77}$Se2B sample was done by manual optimization (see "Methods" section for details) and yielded principal $^{77}$Se hyperfine coupling values of $A_x = 3$ MHz, $A_y = 10.5$ MHz and $A_z = 0$ MHz ($a_{iso}$ ($^{77}$Se) ~ 4 MHz) (grey shaded dotted traces in Fig. 5). Of these values, only $A_y$ can be trusted, as

$B_0 = 560$ mT corresponds to the effective $g_y$ principal value of the $\lambda_2$ species. Variations of $A_x$ and $A_z$, especially at higher magnetic fields, do not affect the quality of the simulations, so both values are undefined.

## Discussion
### Regularization versus grid-of-error approach
For the analysis of the complex cw-EPR spectra, two model-free methods, the grid-of-errors method[37] and the regularization method,

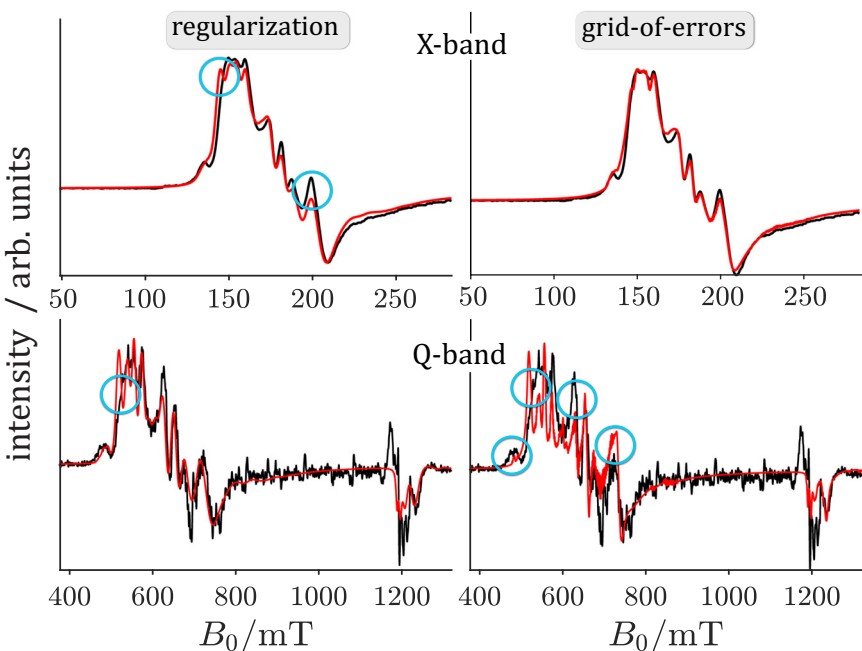

**Fig. 7 | Comparison of results using the regularization (left) or the grid-of-error (right) method.** The X-band cw-EPR spectrum (**upper panel**) and the pseudo-modulated Q-band pulse EPR spectrum (**lower panel**) of sample Av1-Se2B-1 were used as example spectra. Areas where the respective methods do not reproduce the experimental data well are highlighted as blue circles.

were chosen to identify and analyze the individual spin species. The former method has been successfully applied to a high-spin Fe-EDTA complex[37,45]. An accurate $|E/D|$ value is necessary for both methods, as only then the computed rhombicity values can be converted into a correct effective $\boldsymbol{g}$-tensor (see Supplementary Methods). In the Fe-EDTA system, $|D|$ is not significantly larger than the electron-Zeeman splitting in X-band, therefore measurements at several magnetic field strengths and simultaneous evaluation of all spectra with the grid-of-errors approach lead to accurate $|D|$ values[45]. In the FeMo cofactor, $|D|$ ($\approx 180$ GHz[19]) is much larger than the electron Zeeman splitting in X-band ($\approx 10$ GHz) and, therefore, $|D|$ can only be precisely determined at frequencies above $|2D|$, or by performing a frequency sweep experiment at different magnetic fields[46]. Such experiments are quite difficult to perform in terms of sample size and experimental conditions; however, the qualitative analysis performed in this study showed that $|D|$ can be safely assumed unchanged in all samples (Supplementary Fig. 18).

For analysis of the experimental FeMo cofactor spectra, a lwpp of 2.5 mT was determined for all samples at 5 K from a lwpp analysis (determination of the minimum in a $\rho$(lwpp) versus lwpp plot) and was used in the regularization method (Supplementary Fig. 19). The lwpp was used as a second independent parameter in the grid-of-error approach; this may be advantageous if the lwpp differs from species to species. In the Se-incorporated samples only $\lambda_4$ showed a lwpp of more than 5 mT, while the linewidths of the other species matched the value of 2.5 mT quite well (Supplementary Fig. 25).

To best analyze the quality and robustness of both methods, the X-band cw-EPR spectrum and the pseudo-modulated and $\tau$-averaged Q-band pulse EPR spectrum of sample Av1-Se2B-1 (Fig. 6) were analyzed by both methods in X-band, and the results were compared with

Both analytical methods were first tested and compared using three model systems. A fixed intrinsic lineshape lwpp of 1 mT was used, thus the only variable parameter in these simulations was the rhombicity $\lambda$. Comparison of the calculated and simulated spectra showed that the grid-of-errors approach, in particular in the case of low S/N, gave inferior results in comparison to the regularization method, which consistently performed exceedingly well (Supplementary Methods).

the experimental data in X- and Q-band (Fig. 7). Because lwpp is a second independent parameter in the grid-of-errors approach, slightly better results are obtained in the simulation of X-band cw-EPR spectra than using the regularization (Fig. 7, upper panels). On the other hand, the lwpp parameter is slightly overestimated by the grid-of-errors method (Fig. 7, lower panels), which lowers the quality of these results in Q-band. Overall, spectral simulations obtained from either method are of excellent quality and show only minor deviations from the experimental data.

### Regularization of published data

In addition to the three model systems, previously published experimental data were analyzed by regularization to evaluate the reliability of this method. First, EPR data sets from the Hoffman group of the freeze-quenched turnover intermediates of a nitrogenase complex, with and without blue light irradiation, were used[14]. The cw-EPR spectra were initially simulated manually, obtaining three species with different $g$-values. Besides the resting state signal, the two additional species were assigned to two different hydride intermediate signals (1b and 1c) of the $E_2$ state ($E_2$(2H)). Low-temperature blue light irradiation leads to a change in the population of 1b and 1c. The results of the regularization method (Supplementary Fig. 14, left and central panels) fit very well with the experimental cw-EPR spectra, and the respective probability distribution (Supplementary Fig. 14, right) is in strong agreement with the analysis performed in[14]: The intensity ratio of the two $\lambda$ values that represent signals 1b and 1c changes in the direction of state 1c after blue light irradiation.

A second previously published dataset[47] was also analyzed by the regularization method. Here, the EPR data of the MoFe-protein from Av1 and *Clostridium pasteurianum* (Cp1) show the protonated state $E_0(H)^+$ in addition to the resting state signal $E_0$ at low pH values. An additional signal with a $\lambda$ value of 0.11 (Av1) and 0.85 (Cp1) was detected by regularization, which is in line with the previous analysis (Supplementary Fig. 15). Only the spectrum of Av1 at pH 5 shows small differences between experiment and regularization, this may be caused by an additional signal at $g \approx 2$ and baseline artifacts. Both

examples clearly show that our model-free analysis is capable of analyzing various data sets accurately.

This detailed investigation hence demonstrates that regularization is a powerful and fast approach to simulate EPR spectra that are either dominated by only one statistically-distributed parameter (in this case, $\lambda$), or depend only on a second, non-dominant parameter (in this case, lwpp). To further improve the accuracy of the distribution $P(\lambda)$, the samples could be measured in several frequency bands[37], and evaluated using a global regularization analogous to the analysis of DEER data sets[34]. In summary, we believe that this method is more applicable to a high-spin EPR system than any simple strain models since it provides faster, better, and model-free results for systems with many states and, therefore, many parameters.

### EPR analysis and assignment of Se-incorporated samples

Pulsed- and cw-EPR experiments revealed that all species contain a total spin of 3/2, and all Se-species ($\lambda_{1,3-5}$) relax faster than the FeMo resting state ($\lambda_2$). 3P-ESEEM experiments of sample Av1-Se$^{77}$Se, which is labeled only at position 2B, confirm that Se is incorporated into the cofactor as its presence leads to additional hyperfine couplings. The same interpretation can be assumed for sample Av1-$^{33}$Se, although only spectroscopic, no crystallographic confirmation is available for this sample[28]. Spectral simulation revealed that the $A_y$ value of the Se hyperfine coupling is about 10.5 MHz, while the other two principal values $A_x$ and $A_z$ have to be treated with caution as their values only moderately influence the quality of the simulations. In addition to dead-time artifacts and cross suppression, the different hyperfine couplings of the individual spin species also impede unambiguous simulation results. The two S-labeled samples (Av1-S and Av1-$^{33}$S) were generated under turnover conditions in the presence of KSCN without N$_2$. Sample Av1-$^{33}$S exhibits additional hyperfine and quadrupole couplings with a strength of only a few MHz, which originate from one $^{33}$S, and demonstrate that S exchange occurs even in the absence of N$_2$. We note that ENDOR experiments using uniformly $^{33}$S-labeled nitrogenase have already been conducted. $^{33}$S hyperfine couplings between −10 MHz and −16 MHz, including a quadrupole coupling of ~1 MHz, have been reported, but no specific S atom could be assigned[19]. In summary, additional hyperfine couplings in the 3P-ESEEEM spectra can be simulated by only one additional isotope ($^{33}$S or $^{77}$Se).

All Se-incorporated samples contain four additional spin species ($\lambda_{1,3-5}$), indicating that Se-exchange is possible under all experimental conditions studied (see Enzyme assays section in the Method section), most likely with yields above 90% at position 2B[28,30]. Results from regularization (and from the grid-of-errors approach) show that regardless of the expected distribution of Se within the sulfur belt, the cw-EPR spectra always show similar rhombicity distributions and vary only in their probability intensities (Fig. 3D-H and Table 1). As crystallographic studies confirm different labeling pattern[28], it is possible that only the exchange at position 2B is detected spectroscopically and that additional Se exchange at positions 3 A and 5 A does not involve further changes in the electronic structure of the cluster. Note that Henthorn and colleagues carried out cw-EPR measurements with similarly prepared samples and detected no relevant changes in the EPR signals (Fig. S2 in Ref. 30). This does not contradict our results, as a closer look at their cw-EPR spectra reveals some additional low-intensity signals. Besides slightly different sample preparations, the reason could be the increased temperature of their measurements (10 K versus 5 K). Comparable cw-EPR measurements at 12 K support this interpretation: due to the short relaxation times of the FeMo cofactor, only a significantly broadened Se-pattern of low intensity can be detected (Supplementary Fig. 22).

To gain insights into the nature of the four Se species, published EPR parameters of freeze-quenched reaction intermediates of the

**Table 2 | Assignment of λ values to intermediate states of the FeMo cofactor**

| | | $\lambda_1$ | $\lambda_2$ | $\lambda_3$ | $\lambda_4$ | $\lambda_5$ |
|---|---|---|---|---|---|---|
| Experimental values | | 0.033 | 0.056 | 0.082 | 0.116 | ≈0.2 |
| Literature values[14,22,23,47,49] | $g'_y{}^{1/2}$ | ≈4.25 | | ≈4.5 | ≈4.7 | |
| | $g'_x{}^{1/2}$ | ≈3.75 | | ≈3.57 | ≈3.3 | |
| | $g'_z{}^{1/2}$ | ≈2.00 | | ≈1.99 | ≈2.00 | |
| | $\lambda$ | ≈0.04 | | ≈0.08 | ≈0.12 | |
| Literature assignment | | E$_2$(2H) | E$_0$ | E$_2$ | E$_2$(2H) and/ or E$_0$(H)$^+$ | ? |
| Assignment of this study | | E$_2$(2H) | E$_0$ | unknown or E$_2$ | E$_2$(2H) | ? |

FeMo cofactor were extracted and compare with our values[14,22,23,48]. Two identified $S = 3/2$ spin states (1b and 1c)[22,48] have been previously assigned to hydride isomers of state E$_2$(2H)[14]. The effective $g'$-factors of these species were extracted, and by using the equations $\lambda = \frac{2(\Delta g)}{3(\Delta g)^2 - 1}$ and $\Delta g = \frac{g'_y{}^{1/2} + g'_x{}^{1/2}}{g'_y{}^{1/2} - g'_x{}^{1/2}}$ (derived from Eq. 3 in the Supplementary Informations) and assuming $g_x = g_y$, the rhombicity values of these states may be calculated as $\lambda_{1b} \approx 0.04$ and $\lambda_{1c} \approx 0.114$, respectively. Other studies assigned a $S = 3/2$ spin state with a $\lambda$ value of about 0.12 to the protonated resting state E$_0$(H$^+$)[47,49], and a photoinduced state with a $\lambda$ value of about 0.08 was very recently assigned to the (protonated) E$_2$ state[49]. Moreover, in freeze-quench experiments during turnover using an α−70Ile variant of Av1, a rhomboid signal ($\lambda = 0.24$) was assigned to state E$_2$[50]. This state can be excluded to be present in any of our samples as $\lambda = 0.24$ is well above the $\lambda$ values observed in our spectra. The fact that a variant was used could explain the different $\lambda$ values of the study and the one by Chica and coworkers[49]. An $S = 1/2$ signal with a rhombic $\boldsymbol{g}$-tensor in the region of $g \approx 2$ that was assigned to state E$_4$[15,50] is again not observed in any of our Se-incorporated Av1 spectra.

These literature values and the $\lambda$ values determined in this study are summarized in Table 2 and allow a comparison: Species $\lambda_1$ has very similar values to the assigned state E$_2$(2H), species $\lambda_3$ to the assigned (hydrogenated) state E$_2$, and species $\lambda_4$ to the assigned states E$_2$(2H) or E$_0$(H$^+$). Species $\lambda_5$ has never been observed in any other EPR experiment yet. Despite geometric distortions, $\lambda_5$ could represent another hydride isomer of E$_2$ or of any other higher state, as long as the total spin is $S = 3/2$.

The combination of the literature comparison, the analysis of the species distribution of sample Av1-S-remigration, and the results of the experiments with blue-light irradiation together support a more definite assignment of the different Se-species: Exchange of Se back to S, which was the rationale behind preparing sample Av1-S-remigration, does lead to a reduction of the states $\lambda_1$ and $\lambda_4$, but besides the ground state ($\lambda_2$), state $\lambda_3$ persist even after prolonged reaction cycles. The light-irradiation experiments show very similar results: The probability of state $\lambda_3$ (and $\lambda_2$) does not change in response to light. On the other hand, $\lambda_1$ and $\lambda_4$ respond reversibly to blue light, but whether a conversion of the hydrides upon light illumination really takes place, or only partial photolysis, needs to be clarified by further experiments.

Therefore, states $\lambda_1$ and $\lambda_4$, whose $g$-factors are very similar to those reported by[14], and were referred to as states 1b and 1c, are most likely two different hydride isomers of state E$_2$. State $\lambda_3$, on the other hand, is irreversibly formed, representing a non-productive state that cannot be re-exchanged. It could be a metastable protonated E$_0$ state or a metastable additional hydride, which is irreversible due to the different p$K_a$ value of the Se-FeMo cofactor (see below).

If the published assignments of the intermediate stages were not to be trusted, could, in principle, all Se species originate from geometric distortions? A number of findings speak against such an interpretation: First, it is highly unlikely that the $g$-values of unlabeled FeMo intermediates and of geometrically distorted Se-FeMo cofactors are very similar (Table 2). Second, if the individual Se-species would result from a simple geometric distortion of the FeMo cofactor by incorporation of the larger Se atom, either none or all of the Se-species would be re-exchanged by S in the Av1-S-remigration sample. Third, geometric distortions would lead to different $\lambda$-distributions of samples with Se-exchange at position 2B (samples Av1-Se2B-1 and Av1-Se2B-lowflux) compared to samples with equal labeling of the sulfur belt (sample Av1-Se-C$_2$H$_2$), yet, our distributions do not show differences between the samples. In this context, it must be noted that the ground state values of the Se-FeMo and FeMo cofactors differ slightly (green dashed lines in Fig. 3), and hence potential differences in geometry may have a minor influence on the $\lambda$ values.

A further important aspect in the discussion of the individual Se species is the decreased signal intensity of the Se-labeled samples compared to the S- or unlabeled samples: 40% of the FeMo cofactors are in an EPR silent state, which confirms that EPR inactive intermediate states of the FeMo cluster like E$_1$ or E$_3$ are also stabilized by the Se-incorporation method. As the S-labeled FeMo cofactors have the same cw-EPR intensity as the unlabeled cofactor, the S-to-S exchange does not stabilize any intermediate states.

## Mechanistic insights

The question remains as to why intermediate states are stabilized by the incorporation of Se. Se has a higher polarizability compared to S, and the Se-H group has a lower p$K_a$ value compared to the S-H group[51], while serving as a structural surrogate for S in iron-sulfur clusters[52]. Moreover, calculations on Se (or S) metal model complexes discovered that the substitution of S with Se leads to a reduction of the ligand field strength and can additionally affect the energy of the electronic states[53]. These differences could lead to an equilibrium shift of the overall reaction upon Se substitution within the FeMo cofactor and favor side reactions to early intermediate states (E$_3$, E$_2$, and E$_1$) accompanied by the release of H$_2$. No states higher than E$_2$ are observed, suggesting that the incorporation of Se into the FeMo cluster has to occur very early in the reaction scheme. The incorporation of Se into the 2B position of the FeMo cofactor could be accomplished via different reaction pathways[54,55]. Our results support mechanisms that include protonation steps, as direct Se labeling would likely not result in as many different hydride isomers.

Our results demonstrate that Se incorporation leads to the stabilization of different intermediate states containing different electronic structures. These differences could be due to changes in the effective oxidation states of the Fe atoms in the FeMo cofactor, whereby the total spin of $S = 3/2$ must be maintained. X-ray spectroscopy with a Se-labeled FeMo cofactor showed that position 2B and positions 3 A/5 A are electronically different[30]. It was observed that the two iron atoms (Fe2/Fe6) that bind the Se at the 2B position show a local oxidized character, whereas the iron nuclei which bind to the Se atoms at positions 3 A/5 A are rather reduced. It was also noted that both the incorporation of Se and hydrogen bonds affect the effective oxidation state and the electronic structure[30]. It is important to recognize that stable forms of the VFe-protein and FeFe-protein related to turnover intermediates have been previously reported with N/O incorporated at a belt sulfur position[29,56]. These observations provide precedent for the observations described here that intermediates states of the cofactor may be stabilized by the replacement of a belt sulfur by a non-sulfur ligand.

Can the additional hydrides of the E$_2$(2H) intermediate states be detected and characterized by EPR spectroscopy? Basically, additional protons show up in 3P-ESEEM spectra as additional signals around the proton Larmor frequency. Insets in Fig. 6B show these regions magnified for samples Av1-WT and Av1-Se2B-1. The proton hyperfine couplings of the Se-labeled sample (red) show a broadening compared to the unlabeled sample (black), and a weak splitting can be observed in the spectrum at the magnetic-field position 740 mT (see also Supplementary Fig. 31). Both of these indicate additional proton hyperfine couplings. ENDOR studies on the resting-state FeMo cofactor as well as on the CO-labeled cofactor, have shown that the hyperfine couplings of the surrounding protons have only the strength of only a few MHz[19,57]. It is, therefore, likely that any additional hyperfine couplings are only hardly visible in the 3P-ESEEM spectra due to fast relaxation times, low modulation depth, and cross-suppression effects and are masked by the linewidth. It can still be concluded that the incorporation of Se leads to a broadened proton hyperfine signal pattern that most likely originates from additional protons attached to the Se-FeMo cofactor. Again, ENDOR spectroscopy at about 2 K could be helpful to further characterize these additional protons, in particular as the signals from species $\lambda_{1-5}$ are at least partially spectrally separated; a combination of blue-light illumination and orientation selection can further reduce the number of Se-species and enable unequivocal assignment.

## Application of regularization in EPR spectroscopy

In this study, various Se incorporation experiments into the catalytically active FeMo cofactor of a nitrogenase were investigated by EPR spectroscopy, as the property of such labels, e.g., their different reactivity, are far from being fully understood[28]. cw-EPR spectra of Se-incorporated samples showed complex signal patterns compared to unlabeled samples. Using two different analysis methods, the Tikhonov regularization and the grid-of-errors approach, five different electronic states could be identified, one of which is assigned to the E$_0$ ground state, and the others to (protonated) intermediate states (E$_0$(H$^+$) and E$_2$(2H), see Table 2) of the cofactor. These experiments confirmed the incorporation of at least one Se (or S) atom under turnover conditions. As only early intermediates of the LT scheme were detected, the opening and incorporation of Se (and presumably also of other substrates) is very likely to proceed in the first steps of the reaction. It is also important to mention that 40% of the FeMo cofactors are in an EPR-silent state after Se-incorporation. Even if state E$_1$ is the most probable of these states, higher odd states are also in principle possible. The result that under the selected experimental conditions a defined incorporation of Se or S takes place and reaction intermediates can be stabilized without effort offers great potential with respect to further investigations using different molecular spectroscopy methods.

Even though regularization methods for the robust solution of ill-posed problems have been available for some time[58], their application in the life sciences has only recently become popular. In addition to the analysis of spectroscopic data[59], more and more large data sets from genome research have been analyzed successfully, particularly in combination with machine learning[60]. The results presented here provide another successful application of the regularization method; in this case complex EPR spectra with multiple species could be analyzed; this approach may be applied to other systems that contain several overlapping high-spin ($S = 3/2, 5/2, ....$) species. It should be noted in this context that the regularization method can easily be extended to systems with half-integer spin higher than $S = 3/2$, and that the cw-EPR signals of high-spin systems, are usually dominated by only one parameter, the rhombicity, while other parameters, such as the linewidths of the individual EPR signals, have similar values.

**Table 3 | MoFe-protein samples and modifications used in this study. All turnover assays are performed without nitrogen**

| Sample | Abbreviation | Sample condition | Labeling (based on [28,30]) | Concentration |
|---|---|---|---|---|
| *Azotobacter vinelandii* MoFe protein (Av1) wild type – resting state | Av1-WT | 100% Av1 | -- | ≈48 mg/mL |
| Av1 wild type (turnover) | Av1-Se2B-1 | 10 mM KSeCN was added to the turnover assay (Av2/Av1 ratio = 2). | Se (position 2B) | ≈73 mg/mL |
| Av1 wild type (turnover) | Av1-$^{77}$Se2B | 10 mM K$^{77}$SeCN was added to the turnover assay (Av2/Av1 ratio = 2). | $^{77}$Se (position 2B) | ≈74 mg/mL |
| Av1 wild type (turnover) | Av1-Se-low | 0.25 mM KSeCN was added to the turnover assay (Av2/Av1 ratio = 2). | Se (predominantly at position 2B). | ≈70 mg/mL |
| Av1 wild type (turnover) | Av1-$^{33}$S | 10 mM K$^{33}$SCN was added to the turnover assay (Av2/Av1 ratio = 2). | $^{33}$S (position 2B) | ≈40 mg/mL |
| Av1 wild type (turnover) | Av1-Se-C$_2$H$_2$ | sample Av1-Se2B-1 was used for a second turnover assay, which was quenched with 10 mM KSeCN after a reaction time of 5 min. | Se (positions 2B, 3 A and 5 A) | ≈70 mg/mL |
| Av1 wild type (turnover) | Av1-Se2B-lowflux | 15 mM KSeCN was added to the turnover assay (Av2/Av1 ratio = 1.5). | Se (position 2B) | ≈57 mg/mL |
| Av1 wild type (turnover) | Av1-S | 22.5 mM KSCN was added to the turnover assay (Av2/Av1 ratio = 1.5). | S (position 2B) | ≈40 mg/mL |
| Av1 wild type (prolonged turnover) | Av1-S-remigration | Sample Av1-Se2B-lowflux was used. A second turnover assay was started with an Av2/Av1 ratio = 4. The assay proceeded for ≈1h. | Se is expected to be replaced by S again. | ≈46 mg/mL |

In the field of transition metal cofactors, iron cofactor enzymes, such as the large family of non-heme hydroxylase, have a potential application as many mechanistic questions are still open[61,62]. In addition, overlapping species of complex manganese cofactors, such as the water-oxidizing complex[63] and transition metal heme complexes[64,65] can be deconvoluted by regularization. Looking beyond transition metal cofactors, organic functional materials such as molecular magnets[66] might represent potential applications.

## Methods

### Sample preparation

The MoFe-protein and Fe-protein (designated as Av1 and Av2, respectively) were isolated as follows: *Azotobacter vinelandii* (Lipman, 1903) was grown in modified Burk's medium (pH 7.5) and bubbled with air. The pre-culture medium, including 10 mM NH$_4$Cl as nitrogen source, was inoculated with *Azotobacter* glycerol strains (1:1 v/v cell solution (OD$_{600}$ = 3–4) with 80% aqueous glycerol) and grown at 30 °C with shaking at 180 rpm. Main cultures (60 L) were complemented with 1.3 mM NH$_4$Cl resulting in short-term repression of nitrogenase gene expression, reversible upon ammonium depletion. The main culture was grown in a 60 L bioreactor at 30 °C with stirring of 180 rpm, and air bubbled through media at 50 L/min. Cells were harvested by centrifugation at an optical density (OD$_{600}$) of 2.0.

All protein-handling steps were performed anaerobically. Buffers were degassed using Ar-gas (vacuum-Ar purge cycles) followed by addition of 5 mM Na$_2$S$_2$O$_4$ at pH 7.5. Cells were ruptured in a high-pressure homogenizer (Emulsiflex C5, Avestin) under Ar atmosphere. The cell lysate was centrifuged at 18,900 × g for 30 min and the supernatant was loaded onto a HiTrap Q anion exchange column (GE Healthcare) pre-equilibrated with a 50 mM Tris/HCl (pH 7.5), 100 mM NaCl buffer. The MoFe protein was eluted with a linear NaCl gradient at ~350 mM, and the Fe protein was eluted at ~475 mM. After collection, each protein sample was concentrated and loaded onto a size exclusion column (S200, 26/60, GE Healthcare) equilibrated with 50 mM Tris/Cl (pH = 7.5), 200 mM NaCl buffer. Pure MoFe protein was concentrated to ~60 mg mL$^{-1}$ using an Amicon concentrator (100,000 kDa MWCO, Millipore Ultracell) under 5 bar Ar pressure. Fe Protein was concentrated to ~50 mg mL$^{-1}$ using an Amicon concentrator (30 kDa MWCO, Millipore Ultracell) under 5 bar Ar pressure. Nitrogenase activity was assayed by monitoring acetylene reduction.

### Enzyme assays

Turnover assays for Av1 and Av2 were prepared in a buffer containing 50 mM Tris-HCl (pH 7.5), 200 mM NaCl, 5 mM Na$_2$S$_2$O$_4$ and supplemented with 20 mM creatine phosphate, 5 mM ATP, 5 mM MgCl$_2$, 25 units/mL phosphocreatine kinase and 25 mM Na$_2$S$_2$O$_4$ (in 50 mM Tris-HCl, pH 7.5 and 200 mM NaCl)[28]. All samples except for the Av1-Se-C$_2$H$_2$ sample were kept under an argon/H$_2$ atmosphere, and the indicated amounts of KSCN, K$^{33}$SCN, KSeCN, or K$^{77}$SeCN were added to the reaction (see Table 3). C$_2$H$_2$ was used as substrate in the Av1-Se- C$_2$H$_2$ sample. Afterward, the Av2 protein and remaining SCN$^-$ or SeCN$^-$ were removed by three rounds of sample concentration and dilution with a 100-kDa molecular weight cut-off ultrafiltration device (Vivaspin, Sartorius). An additional desalting step (Sephadex G25, GE Healthcare) was applied with samples Av1-WT, Av1-$^{33}$S, Av1-Se2B-1, Av1-Se-C$_2$H$_2$, Av1-Se-low and Av1-$^{77}$Se2B. Sample concentrations were determined by absorbance at 410 nm[28]; relative EPR signal intensities were determined by double-integration of the respective X-band cw-EPR spectra.

### cw-EPR experiments

X-band cw-EPR experiments were performed using Bruker E500 or E580 spectrometers in combination with Bruker resonators

(4122SHQE or 4119HS-W1) combined with an Oxford ESR900 helium gas flow cryostat. Power-sweep experiments were done at 5 K, a microwave frequency of 9.39 GHz, a modulation amplitude of 0.6 mT, and a conversion time of 165.25 ms. For testing the relaxation behavior of the individual samples, cw-EPR spectra at different microwave powers (from 0.025 to 39.4 mW at the E500, or from 0.377 to 37.7 mW at the E580) were recorded. Temperature-dependent experiments were recorded at 6, 9, or 12 K using a microwave power of 0.095 mW, a modulation amplitude of 0.6 mT, and a conversion time of 165.25 ms.

### Light induced cw-EPR experiments

Similar to the protocol described in[14], two samples, Av1-Se2B-1 and Av1-Se-low, were illuminated inside the cooled cavity (Bruker 4119HS-W1) in combination with the cryostat (Oxford ESR900) for about 10 min using a blue-light LED (100 mW, Schott KL 2500). The cw-EPR experiments were performed at 6 K at 9.38 GHz by using microwave power of 3.77 mW, a conversion time of 160 ms, and a modulation amplitude of 0.6 mT. The cryogen annealing was done by keeping the samples in a cryogen-solution (isopropanol-liquid nitrogen) at about 150 K for some hours. Additionally, the samples were stored for 16 h in liquid nitrogen.

### Pulse EPR experiments

Pulse Q-Band EPR experiments were performed using a Bruker E580 spectrometer in combination with a Bruker EN 5107D2-flexline resonator immersed in an Oxford CF935 helium gas-flow cryostat. All experiments were carried out at a microwave frequency of 33.8 GHz at 4.5 K. Unless noted otherwise, a video gain setting of 200 MHz was used.

### Longitudinal transient nutation experiments

Experimental conditions: pulse length $\pi/2 = 10$ ns, nutation step width 4 ns, $\tau = 110$ ns, $T = 600$ ns, and a shot repetition time of 51 μs. A 4-step phase cycle was used. The spectra were measured in steps of 10 mT. As the nutation frequency depends on the local microwave magnetic field strength $B_1$, all frequency axes were normalized to the nutation frequency measured with a coal reference sample (Bruker). This standardization makes the frequency axis essentially independent of spectrometer-specific settings such as microwave power or the resonator quality. The nutation signals were processed as follows: After subtraction of a polynomial baseline, a Hamming window function and a zero filling with a fill factor of 4 were applied. Finally, an FFT was performed.

### Inversion recovery experiments

Experimental conditions: pulse lengths $\pi/2 = 12$ ns, $\tau = 100$ ns, $T_{start} = 400$ ns, $T$-steps = 80 ns, and a shot repetition time of 100 μs. The video gain was set to 20 MHz. The spectra were measured in steps of 3 mT. From each spectrum, the resonator background was subtracted. Exponential fit functions were used to determine $T_1^{eff}$.

### 2-pulse ESEEM versus $B_0$ experiments

Experimental conditions: pulse length $\pi/2 = 12$ ns, $\tau_{start} = 100$ ns, $\tau$-steps = 4 ns with 40 steps and a shot repetition time of 20 μs. The spectra were measured in steps of 0.3253 mT. The resonator background was subtracted from each spectrum. Pseudo modulation was performed using a modulation amplitude of 1.0 mT and a binominal smoothing with 4 smoothing points.

For determining $T_M^{eff}$, modified experimental conditions were used: pulse length $\pi/2 = 12$ ns, $\tau_{start} = 100$ ns, $\tau$-steps = 4 ns with 500 steps and a shot repetition time of 50 μs. A 16-step phase cycle was used. The spectra were measured in steps of 3.0 mT. Exponential fit functions were used for analysis.

### 3-pulse ESEEM experiments

Experimental conditions: pulse lengths $\pi/2 = 10$ ns, $\tau = 90$ ns, $T_{start} = 100$ ns, $T$-steps = 8 ns with 750 steps, shot repetition time 70 μs. The spectra were measured in steps of 10 mT. A 4-step phase cycling was used. Spectra have been processed as follows: The phase of the time domains have been optimized, a mono- or bi-exponential background function has been subtracted, a Hamming window function has been applied, a zero-filling factor of 4 has been used, and finally, a cross-term-averaged FFT was applied.

### Data Analysis

Spectral simulations of cw-EPR spectra were carried out using the Matlab (The MathWorks, Natick, MA) package EasySpin[67] with its pepper simulation routine; spectral analysis was done using self-written Matlab scripts. The regularization and the grid-of-errors method were implemented as Matlab scripts (for details, see Supplementary Methods). The regularization results were analyzed using a multi-Gaussian approach described in the Supplementary Fig. 26.

3P-ESEEM simulations were carried out using the EasySpin algorithm saffron[68]. Pseudo-nuclear and effective hyperfine couplings were included by calculating with a total electron spin quantum number of $S = 3/2$ and zero-field coupling $D = 180$ GHz. ESEEM signals of the two nitrogen atoms were simulated using literature parameters: $A(N1) = [1.02\ 0.98\ 1.14]$ MHz, $Q(N1) = 2.17$ MHz/$\eta$(N1) = 0.6, and $A(N2) = [0.5\ 0.4\ 0.4]$ MHz, $Q(N2) = 3.5$ MHz/$\eta$(N2) = 0.35; Euler angles of 60°, 20°, 0° between the **g**- and quadrupolar-tensor for the second nucleus axis were used[44].

Spectral simulations of ESEEM signals of sample Av1-$^{77}$Se2B were performed as follows: Using the determined $^{14}$N hyperfine couplings and assuming one $^{77}$Se nucleus, the analysis of the spectral pattern in the Av1-$^{77}$Se2B sample was done by manual optimization. For a one-to-one simulation, different rhombicity ($\lambda$) values that affect both the effective hyperfine couplings and the pseudonuclear g-factors were taken into account. These effects scale with the magnitude of the hyperfine couplings and, thus, alter the effective nuclear Larmor frequency and the effective hyperfine couplings. For this reason, simulations were performed for each $\lambda$ value individually. The simulations were done between $0 \le \lambda \le 1/3$ in 167 steps. For each $\lambda$ value the 3P-ESEEM spectra $S(\lambda)$ were calculated and weighted by the probability-distribution $P(\lambda)$ obtained from regularization. The total spectrum was obtained by: $S = \sum_\lambda S(\lambda)P(\lambda)$. Only the lower Kramers doublet was considered.

### Runtime estimation of the regularization and grid-of-errors methods

Using a standard desktop PC (Intel Core i5-4590 CPU @ 3.3 GHz) with Matlab 2019a and EasySpin 5.2.25, the calculation of the kernels (667 $\lambda$-steps with $0 \le \lambda \le 1/3$, 18 intrinsic lineshape-steps (lwpp) with 0.5 mT steps, and an angle step width of 0.5° for calculation of a powder spectrum) required about 12 hours. The regularization itself, using 27 $\alpha$-values and 18 lwpp points required a compute time of approximately 150 seconds. It is therefore time-saving to calculate the kernel once per series of spectra. On the other hand, the calculation of the grid (223 $\lambda$-steps with $0 \le \lambda \le 1/3$, 249 lwpp-steps 0 mT $\le$ lwpp $\le$ 25 mT, and an angle step width of 0.5° for calculation of a powder spectrum) required about 56 hours. The grid-of-errors optimization itself required only about 300 seconds. The comparison of the compute times clearly shows that the regularization method requires less computing time and should therefore be preferred over the grid-of-errors method if the prerequisites for regularization are fulfilled (see below). The reduction of computation time is mainly due to the lower required number of steps in the second parameter dimension (here: lwpp). By choosing an identical

# Article

number of steps for $\lambda$ and lwpp, the compute times for both methods are very similar.

## Reporting summary

Further information on research design is available in the Nature Portfolio Reporting Summary linked to this article.

## Data availability

All data supporting the findings of this study are available with the paper and its supplementary information files. The raw data can be downloaded from the website: https://freidok.uni-freiburg.de/data/246337. Figure 2 is made from the folder Powersweeps_E580, Fig. 5 is made from the folders 2P_ESEEM and 3P_ESEEM, and Fig. 6 is made from the folder Light_Induced. A detailed description of which data was used for which figure can be downloaded from the website: https://freidok.uni-freiburg.de/data/246957. The PDB entry 4TKU can be downloaded from: https://www.rcsb.org/structure/4TKU. Source data are provided as a Source Data file. Source data are provided in this paper.

## Code availability

All code used in this work can be downloaded from the website: https://freidok.uni-freiburg.de/data/246338.

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

## Acknowledgements

We thank Jon Rittle for kindly preparing K77SeCN, and Maximilian Mayländer and Johannes Ruhnke for help with the blue-light EPR experiments. We thank Prof. Dr. Brian Hoffman for providing the EPR data sets of freeze-quenched and light-induced samples. Support from NIH Grant GM045162 and the Howard Hughes Medical Institute to D.C.R. is gratefully acknowledged. O.E. acknowledges support from the German Research Foundation (PP 1927, project ID 311061829, and RTG 1976, project ID 235777276). E.S. acknowledges support by the Hans-Fischer-Gesellschaft. L.H., S.W., and E.S. thank the SIBW/DFG for financing EPR instrumentation that is operated within the MagRes Center of the University of Freiburg.

## Author contributions

L.H., K.P., T.S., O.E., S.W., D.C.R. and E.S. designed the research and conceived the experiments. K.P. and T.S. prepared all samples. L.H. conducted all EPR and ESEEM experiments. L.H. and E.S. analyzed and interpreted the spectroscopic data. L.H. wrote the simulation routines. The figures were generated by L.H. and E.S. The manuscript was written through the contributions of all authors. All authors have reviewed and approved the manuscript.

 

## Funding

## Competing interests

The authors declare no competing interests.
