## [Peer Review File · Nature Communications]

Analysis of early intermediate states of the nitrogenase reaction by regularization of EPR spectraREVIEWER COMMENTS

Reviewer #1 (Remarks to the Author):

In their manuscript "Analysis of early intermediate states of the nitrogenase reaction by Se incorporation and regularization of EPR spectra" the authors create nitrogenase samples with different chalcogen incorporations into the FeMo cofactor. These contain ^{33}S , and selenium in natural abundance or isotopically pure ^{77}Se . The EPR spectra change with Se-incorporation. However, all spectra of Se-containing samples are similar leading to the conclusion that the hyperfine couplings to ^{33}S and ^{77}Se are not resolved but that samples including Se have a distribution of rhombicities of the zero-field splitting tensor belonging to the $S=3/2$ paramagnetic centre are present. As the spectral differences originate from a probability distribution of a single parameter that authors take inspiration from processing of DEER distance distributions and apply Tikhonov regularisation to stabilize the solution of the ill-posed inverse problem. This is benchmarked with simulated data and compared to an established grid-of-error approach. Overall the new methodology is presented very convincingly. From an EPR spectroscopists point of view this is a significant achievement. However, this will remain limited to EPR (or other) spectra that can be described by a set of species differing in a single parameter. This seems a quite exotic problem and this reviewer wonders where else would it be useful? The authors make no specific suggestions that indicate wider impact. Furthermore, the conclusions on the structure of the FeMo cofactor and which atoms are replaced by Se seem mainly confirmatory. The discussion is very speculative and even the protonations are tentative and not unambiguously shown even by the ESEEM data. The authors acknowledge all of this and tread carefully in what they state firmly. I wonder if the nitrogenase community feels this is a big step forward for understanding the mechanism? The summary and outlook seem not to mention functional implications.

I would have thought this is of great interest to a more specialised audience interested in unravelling complex multi-component spectra. However, I may be mistaken here and leave this to the other reviewers and the editors to judge.

Reviewer #2 (Remarks to the Author):

The authors report in their manuscript a cw and pulsed EPR study on the FeMo-cluster of the MoFe protein from *Azotobacter vinelandii*. The cw EPR data are analyzed with a regularization method also employed in the analysis of DEER data.

I recommend rejection of the manuscript because the data do not support the interpretation.

1) The authors state that it is not different structures of the FeMo-cluster that yield the different

constituents of the EPR spectra but if it is not structure where do the electronic differences come from? They say the difference is in λ but where does this come from?

2) The authors also do not explain why different electronic structures should be observed with Se but not with S.

3) Most importantly, the authors cannot rule out that the incorporation of Se leads to artificial structures of the cluster. The argument that a crystal structure does not show a difference is not convincing. The most abundant structure may have crystallized but the rest not.

4) I am also not convinced by the regularization method. This method should first be tested on mixtures of known constituents with variations in λ . They need also to provide the rule by which they limited the number of constituents.

Reviewer #3 (Remarks to the Author):

In this well written manuscript Heidinger et al. investigate different Se and S incorporated FeMo cofactors of the catalytic MoFe protein with advanced EPR methods and approaches. Using the challenging methods developed by Rees and coworkers different Se and S isotopes were incorporated into the intricate FeMo cofactor. Samples under various turnover conditions were analyzed. Expertly preparing these difficult samples allowed the authors to stabilize and analyze early intermediate turnover states of the FeMo cofactor.

At the center of the manuscript lays the analysis of the complex EPR spectra with the regularization and the grid-of-errors method. To apply Tikhonov regularization on complex high-spin metal clusters is new to the field. Since these EPR approaches are beyond the expertise of the reviewer, this approach and its conclusions cannot be sufficiently evaluated by the reviewer.

Nonetheless, using Se-labeled FeMo clusters and applying advanced EPR techniques and approaches is innovative and potentially provides new insights into the highly complex and controversial nitrogenase mechanism.

Dear Reviewers,

We are grateful for appraising the manuscript, and for supporting it for publication after major revision. In detail, we changed the following parts of the manuscript that have been commented:

Reviewer#1:

Comments:

- However, this will remain limited to EPR (or other) spectra that can be described by a set of species differing in a single parameter. This seems a quite exotic problem and this reviewer wonders where else would it be useful? The authors make no specific suggestions that indicate wider impact.

→ The reviewer is correct that a general application of the regularization method was not the aim of the manuscript, which focused on the development of this method with one application to the analysis of the Se-FeMo cofactor. However, the regularization method presented here is more broadly applicable as spectral simulations of multiple overlapping high-spin species remain complex. In this context, we would like to note that in such high-spin systems, there is usually only one dominant parameter, the zero-field splitting, while other parameters such as the linewidths of the individual EPR signals are often in the same range. Thus, the advantages of regularization can be exploited in a number of different systems.

To further illustrate the potential of the method, we analyzed two published nitrogenase data sets (Ref. 14 and 49), and these are depicted in Figures SI14 and SI15. In all data sets, the experimental data are very well reproduced, and these analyses are discussed in a paragraph (page 11 and 12).

Although the regularizations presented here were performed for a nitrogenase $S=3/2$ system, the method can be easily extended for higher spin systems. Systems relevant to the method include iron cofactor enzymes, e.g., the large group of non-heme hydroxylases where many mechanistic questions are still unsolved, which become accessible by regularization. Besides iron, various copper cofactors and transition metal heme complexes can also be accessed by regularization. Inorganic functional materials such as molecular magnets also represent potential applications. We have added a section to the manuscript (page 15) presenting some example systems where regularization can be applied.

- I wonder if the nitrogenase community feels this is a big step forward for understanding the mechanism? The summary and outlook seem not to mention functional implications.

→ This work has established that selenated nitrogenase consists of a mixture of states, including the resting state as well as other states that resemble previously characterized intermediates. This conclusion required the development of new methodology to analyze the EPR data. One of the major challenges in studying the nitrogenase mechanism has been how to stabilize and characterize non-resting states. As stabilization of these states was previously only possible under freeze-quench conditions using elaborate protocols, this new method provides a path for future characterization of these species by the nitrogenase community.

Reviewer#2:

Comments:

- The authors state that it is not different structures of the FeMo-cluster that yield the different constituents of the EPR spectra but if it is not structure where do the electronic differences come from? They say the difference is in λ but where does this come from?

→ The entire manuscript is centered around the question of which additional species are present in the EPR spectra of the Se-incorporated FeMo cofactors. The first regularization analysis identified several additional species based on the different EPR parameters, in this case the zero-field splitting λ . The theory underlying the zero-field splitting is explained in detail in the Supporting Information (part A, page 2–3). The concept is that different electronic states of the cofactor have different λ values, so that all states can be clearly distinguished on the basis of this value. These species were assigned by literature comparison and by light-dependent experiments: The states λ_1 and λ_4 are assigned to different $E_2(2H)$ intermediate states, but the nature of the species λ_3 remains unclear, though various possibilities are discussed on page 13-14.

- The authors also do not explain why different electronic structures should be observed with Se but not with S.

→ FeMo intermediates are difficult to stabilize with an unmodified cofactor, and this has only been achieved with freeze-quenching methods, as the S-FeMo intermediates have too short a lifetime at room temperature; first characterizations of freeze-quenched samples have been published recently (e.g. Ref. 14).

With our Se labeling, analogous intermediates (Se-FeMo) can now be produced and analyzed conveniently. On page 14, we describe why the stability of Se-FeMo cofactors is different from that of S-FeMo cofactors. This involves different pK_a values of S and Se and different ligand field strengths of the two atom types. We are convinced that in particular the different pK_a value is responsible for the stabilization of the Se-FeMo intermediates at room temperature. In summary, the S-FeMo intermediates are too short-lived. The incorporation of Se extends the lifetime of the reaction intermediates that they can be studied spectroscopically.

- Most importantly, the authors cannot rule out that the incorporation of Se leads to artificial structures of the cluster. The argument that a crystal structure does not show a difference is not convincing. The most abundant structure may have crystallized but the rest not.

→ We agree with the reviewer that the conformation in a crystal structure does not necessarily correspond to the most common conformation in solution. However, crystallography has been at the forefront of establishing the nitrogenase structure, including both the composition and detailed structure of the resting state FeMo-cofactor. While it has always been a possibility that the crystallized material represented a minority species, the FeMo-cofactor structure has been subsequently confirmed by other techniques. This would not be the case if only a subset of nitrogenase was crystallizing; while we cannot unequivocally disprove that the crystallized protein represents a minority species of the selenated protein, we have characterized the sample under turnover conditions where the Se is eventually displaced by S without a change in crystal packing, suggesting that the Se and S forms are isomorphous (Spatzal et al. eLife 4, e11620 (2015)).

We also note that the ground state λ values of the Se-FeMo and FeMo cofactors differ only slightly (green dashed lines in Figure 3). Since the direct comparison between S-FeMo and Se-FeMo could only be carried out in the E_0 state, the geometric influence of the selenium labeling can be quantified for these samples. The average of λ_2 only changes from 0.0536 to 0.0571 (Table 2). On the one hand, these values show that differences in geometry have an influence on the λ values, but on the other hand it is clear that the changes are very small. Furthermore, the analysis of the published E_2 states (SI Figure 14) shows that the λ values of the S-FeMo cofactors from Ref. 14 differ only minimally from the λ values of the Se-FeMo cofactors. Thus, this analysis further supports our interpretation that the incorporation of Se has only minimal influence on the λ values.

It is important to recognize that stable forms of the VFe-protein and FeFe-protein related to turnover intermediates have been previously reported with N/O incorporated at a belt sulfur position (Sippel et al. Science 359, 1484 (2018) and Trncik et al. Nature Catal. 6, 415 (2023)). These observations provide precedent for the observations described here that intermediates states of the cofactor may be stabilized by the replacement of a belt sulfur by a non-sulfur ligand (in this case, Se).

- I am also not convinced by the regularization method. This method should first be tested on mixtures of known constituents with variations in lambda. They need also to provide the rule by which they limited the number of constituents.

→ An important advantage of the regularization approach is that it is model-free (discussed on pages 11/12). One of the major objectives of this work has been to test the robustness of this method. As a starting point, several model spectra were analyzed using regularization and compared with the already established grid-of-errors method (page 11 / Supporting Information part A). Following up on the reviewer's comment, we additionally analyzed using regularization two previously published data sets, from the Hoffman group and from the Rees group, for comparison with the published analyses. This analysis is described in a new paragraph on page 11. The model-free analysis resulted in an identical number of species in both data sets, whose signal intensities matched the experimental data well. This provides independent validation for the robustness of the regularization approach.

Reviewer#3:

→ no specific questions

In addition to the changes suggested by the reviewers, the manuscript has been carefully checked for exact grammar usage, and some typos have been corrected.

On behalf of all the authors and with best regards,

Yours sincerely,
Erik Schleicher

REVIEWERS' COMMENTS

Reviewer #1 (Remarks to the Author):

The authors have address my main concerns and the analysis of published data widens the scope of the work.

I am still not fully convinced of the significance to the wider readership of Nature Communications but the work is sound and the current version of the manuscript is publishable.

I would strongly urge the authors to make their digital primary data and the tools used for analysis openly available. "Available on reasonable request" is jut not timely anymore (and unfortunately not always honoured).

I have also been asked to comment on whether the concerns of Reviewer #2 have been addressed in the revision

The authors have responded to the points of reviewer 2. I feel – like reviewer 2 – that the point of some intermediates being stabilised by Se substitution could have been clearer in the original submission. The authors did not exactly make great strides to make this clearer. For the small group of researchers investigating nitrogenase with EPR this may be obvious, but for the main readership it would help to be clear and explicit in the discussion of the experiments. The different intermediates are stabilised due to different pKa and electronic energies in the selenated cofactors compared to the native S substitution. The intermediates are assigned based on literature EPR parameters. The rebuttal was much clearer than the manuscript. I suppose the electronic structure changes the g-values but this effect will be dwarfed by the effects of changed rhombicity?

It is claimed that this can be used to study the intermediates prepared by selenation rather than freeze-quench. It there scope to obtain “purer” samples of single intermediates? While it is obviously better than no experimental access to the intermediates it is most often preferable to have as little overlapping species as possible.

I think the addition of literature data should alleviate concerns about the robustness and reproducibility of the regularisation method. I would still urge the authors to make that openly available.

Reviewer#1:

Comments:

- I would strongly urge the authors to make their digital primary data and the tools used for analysis openly available. "Available on reasonable request" is just not timely anymore (and unfortunately not always honoured).
 - The primary data and the source code are now made available to the public (see statements).

- I feel – like reviewer 2 – that the point of some intermediates being stabilised by Se substitution could have been clearer in the original submission. The authors did not exactly make great strides to make this clearer.
 - We have adapted the abstract and the last section of the discussion (page 15) to further emphasize the result that reaction intermediates of the nitrogenase reaction can be easily and effectively generated by selenium labeling.

- I suppose the electronic structure changes the g -values but this effect will be dwarfed by the effects of changed rhombicity?
 - The reviewer raises a good point: We assume that the changes of the g -values are negligible with Se-labeling compared to their changes in rhombicity. The results of the regularization (Figure 3) support this assumption, as otherwise the results would not match the experimental spectra as well, since the g -values are not considered in the regularization. We have added this point to the manuscript (page 6).

- It is claimed that this can be used to study the intermediates prepared by selenation rather than freeze-quench. It there scope to obtain “purer” samples of single intermediates? While it is obviously better than no experimental access to the intermediates it is most often preferable to have as little overlapping species as possible.
 - The samples produced in this study and their experimental conditions provide initial indications of the extent to which the number of intermediates can be controlled. Firstly, the electron flux can be modified by the Av2/Av1 ratio. Two samples with different fluxes, Av1-Se2B-1 and Av1-Se2B-lowflux, were prepared and analysed: Table 1 shows that a lower flux leads to less amounts of intermediates λ_1 and λ_3 , and thus the composition of the intermediates can be actively controlled. Secondly, the Av1-S remigration sample is available, in which Se is first incorporated and subsequently removed. In this case, it is apparent that the λ_1 and λ_4 concentrations are reduced again by the reincorporation of sulphur. Thus, the proportion of the individual intermediates can be varied by adjusting the reaction times of incorporation and remigration.

- I think the addition of literature data should alleviate concerns about the robustness and reproducibility of the regularisation method.
 - Additional literature on the robustness and application of regularization has been added in the last section of the discussion (page 15).